# Effects of Resistance Training in Individuals with Lower Limb Amputation: A Systematic Review

**DOI:** 10.3390/jfmk8010023

**Published:** 2023-02-10

**Authors:** Miguel L. V. V. Rosario, Pablo B. Costa, Anderson L. B. da Silveira, Kairos R. C. Florentino, Gustavo Casimiro-Lopes, Ricardo A. Pimenta, Ingrid Dias, Claudio Melibeu Bentes

**Affiliations:** 1Laboratory of Physiology and Human Performance, Department of Physical Education and Sports, Institute of Education, Federal Rural University of Rio de Janeiro, Seropédica 23890-000, RJ, Brazil; 2Department of Kinesiology, California State University, Fullerton, CA 92831-3547, USA; 3Physiological Sciences Multicenter Graduate Program, Department of Physiological Sciences, Health Sciences Center, Institute of Biological and Health Sciences, Federal Rural University of Rio de Janeiro, Seropédica 23890-000, RJ, Brazil; 4Laboratory of Exercise Pathophysiology, Institute of Physical Education and Sports, Rio de Janeiro State University, Rio de Janeiro 20950-000, RJ, Brazil; 5Department of Physical Education, Federal University of Santa Catarina, Florianópolis 88040-900, SC, Brazil; 6Physical Education Graduate Program, Department of Gymnastics, School of Physical Education and Sports, Federal University of Rio de Janeiro, Rio de Janeiro 21941-599, RJ, Brazil

**Keywords:** strength training, rehabilitation, individuals with disabilities, adaptive physical education, training program

## Abstract

Individuals with lower-limb amputations may have a significant strength deficit. This deficit may be related to the stump length and can lead to changes in gait, reduced energy efficiency, walking resistance, altered joint load, and increased risk of osteoarthritis and chronic low back pain. This systematic review used the Preferred Reporting Items for Systematic Reviews and Meta-Analyzes (PRISMA) guidelines to examine the effects of resistance training in lower limb amputees. Interventions with resistance training and other training methods were sufficient to achieve muscle strength gain in muscles of the lower limbs, improved balance, and improvements in gait pattern and speed when walking. However, it was impossible to determine from the results whether resistance training was mainly responsible for these benefits or even whether the positive effects presented would be observed with only this training method. When combined with other exercises, interventions with resistance training made possible gains for this population. Accordingly, it is noteworthy that the main finding of this systematic review is that the effects may be different according to the level of amputation, with mainly transtibial and transfemoral amputations studied.

## 1. Introduction

The number of people with disabilities is increasing. For instance, the WHO reported that 10% of the world’s population had some type of disability in 1970, whereas approximately one billion people currently live with some type of disability, or approximately 15% of the world’s population (considering the 2010 estimate) [1]. In the United States of America, 1.6 million people were living with the loss of a limb in 2005, which could reach 3.6 million people by 2050 [2]. People with lower-limb amputations may have a significant strength deficit. This deficit may be related to the stump length and can lead to changes in gait, reduced energy efficiency, walking resistance, altered joint load, and increased risk of osteoarthritis and chronic low back pain [3].

Strength imbalances caused by amputation can be relieved through several training programs that help to mitigate the complications caused by amputation in these individuals [3,4]. Resistance training is an exercise mode in which the body’s muscles move against an opposing force. This opposite force can be achieved using equipment, such as weights, elastic bands, machines, or even with the body’s own mass [5]. It is known that people with disabilities suffer from inaccessibility in their daily tasks. One aim of exercise is to facilitate the practitioner’s daily living activities. Physical exercise can be beneficial for people with disabilities, and lack of exercise and disuse of the prosthesis are considered reasons for the strength deficit in amputees [3]. There are several benefits of resistance training for lower limb amputees, such as improved walking, combatting muscle atrophy, bilateral strength deficit reduction, increased strength for stabilization, improved gait, and improved hip strength [4].

In addition to being a public health concern, amputations can generate significant discomfort in the lives of affected individuals. Investigating whether resistance training can help improve the living conditions of these individuals is essential. Furthermore, it is observed that there is a lack of systematic reviews and practical recommendations that contribute to a better understanding of this topic. This review aimed to examine the effects of resistance training in lower limb amputees in order to present practical guidance based on evidence of resistance training protocols in this population.

## 2. Materials and Methods

This systematic review used the Preferred Reporting Items for Systematic Reviews and Meta-Analyzes (PRISMA) guidelines [6]. Articles published until March 2022 were analyzed. The search and selection of articles took place in two stages, from August to October 2021 and from February to March 2022.

### 2.1. Search Procedures and Study Selection

For the searching and selection of articles, the following databases were used: PubMed/Medline, Scopus, Web of Science, VHL, Cochrane, and Embase. The PICO strategy was used, defining the population as lower limb amputees; the intervention as resistance training; the comparison did not apply; and the outcome was any variable related to physical, motor, or physiological capacity on resistance training intervention as a primary or secondary outcome. This strategy defined descriptors in English and was selected from the MeSH vocabulary query. The descriptors “strength training”, “exercise program”, “strengthening program”, “resistance training”, “exercise prescription”, “amputee”, “amputation”, “lower limb”, and “lower extremity” were selected and combined with Booleans, as shown in Table 1.

### 2.2. Eligibility Criteria

The eligibility criteria were defined according to the previously mentioned PICO strategy. Studies were included that (I) were in Portuguese, English, or Spanish, and (II) had the descriptors listed above and titles that made clear the relationship with the theme of resistance training in lower limb amputees. In case there were doubts about the relevance of the article with the theme of the review, the abstract was read, and if the relevance was still in question, the entire article was read to corroborate the decision to include or exclude the study.

Articles were excluded that (I) referred only to upper limb amputations; (II) were literature reviews; (III) were related to resistance training but amputees were not included in their sample; (IV) were related to amputees but not related to resistance training. Several study designs were considered, as the intention was to examine as many studies as possible. The search did not contain a minimum date limit and studies published until March 2022 were considered.

### 2.3. Data Collection Process

A total of 156 articles were found. Table 2 shows the relationship between the articles found and the databases.

After reading the titles, 83 articles were excluded because they did not fit the scope of the research and 36 articles were excluded because they were duplicates. Thirty-eight articles were selected for reading of the abstract. After reading the abstract, 22 were excluded. From the remainder, 16 articles were selected for a full reading. Of the 16 articles, one was excluded due to language criteria, one was a literature review, one did not have resistance training as an intervention, and three did not fit the proposed theme. Therefore, 10 studies were selected for this systematic review. Data from the articles were extracted into an electronic card file. For a complete visualization of the data, the PRISMA flowchart adapted for the context of this work is available below in Figure 1.

### 2.4. Risk of Bias in Individual Studies

The quality of the selected studies was analyzed using the Tool for the assEssment of Study qualiTy and reporting in EXercise (TESTEX) scale [7].

## 3. Results

Through a structured methodology, the search resulted in 10 studies eligible for this systematic review. Table 3 contains the risk of bias within the studies and Table 4 and Table 5 contain theoretical and descriptive data on the articles.

### 3.1. Risk of Bias within Studies

The quality of the studies was analyzed using the TESTEX scale [7]. Table 3 displays the scores of the articles.

### 3.2. Study Characteristics

Table 4 highlights the heterogeneity of these findings. The studies were carried out in several countries, published between 2004 and 2021, and most were published in the last five years [8,9,10,11,12,13]. Among them, 90% of the studies presented amputees of both sexes. The age of the participants varied from childhood to old age. There were different levels of amputation among the participants. However, transtibial and transfemoral amputees were prevalent in the studies. Regarding the methodology, the studies showed plurality but only three (3) studies were randomized clinical trials [11,14,15].

### 3.3. Exercise Approaches and Studies Results

The exercise protocols are described in Table 5. It is noteworthy that this systematic review of the literature considered analyzing resistance training as an intervention; however, 70% of the studies used other techniques in addition to resistance training [8,9,10,11,14,15,16]. The duration of the exercise protocols ranged from 3 to 20 weeks, with the most significant number (30%) of studies lasting 8 weeks [12,14,17]. Weekly exercise frequency ranged from 1 to 3 times a week, with most (50%) studies exercising twice a week [9,12,13,15,17]. One study did not report the weekly exercise frequency [14] and two other studies varied the weekly exercise frequency according to their criteria [11,16]. Studies also reported sessions ranging from 30 min to 2 h for each training session.

## 4. Discussion

This systematic review aimed to examine the effects of resistance training in lower limb amputees. Such studies used various analysis techniques, training protocols, and resistance training as an intervention. It is believed that this systematic review is the first to analyze and compare the effects of resistance training in lower limb amputees.

### 4.1. Studies Qualities

The poor methodological quality of the majority of the studies presented in this review was notorious. However, it is worth noting that only three studies were randomized clinical trials [11,14,15], which can justify the low quality. In addition, one of the randomized clinical trials had reasonably lower quality than the other two [14]. Interestingly, although it was not a randomized clinical trial, one study [17] obtained the highest score in the TESTEX scale.

The failure to present the load used in the exercise protocols, the duration of the sessions, and the conclusion of the intervention, in addition to whether there was any withdrawal or signaling whether or not there were adverse effects may perhaps be the main reasons for the low quality of the studies found, which impacted the quality of this review. Despite the poor methodological quality, the importance of these studies must be considered since this is a topic of interest. Furthermore, the possible difficulty in carrying out more structured studies with this population should be considered, given the context of inaccessibility for people with disabilities and the particularities of each amputation.

### 4.2. Session Duration, Weekly Frequency, and Total Weeks (Months)

The exercise protocols ranged from 3 to 20 weeks, with 8-week interventions being the most common [12,14,17]. Four studies lasted less than eight weeks [9,10,13,14], while six studies lasted more than eight weeks [8,11,12,15,16,17], of which only one lasted longer than 12 weeks, lasting a total of 20 weeks [8]. Some authors consider studies of 8 to 12 weeks to be short, which may not contribute to long-term interventions [4]. It is noteworthy that three of the four studies lasting less than eight weeks did not obtain significant results in at least one outcome measure [10,13,14]. In addition, some studies of more than eight weeks in duration presented in this review also showed little or no significant change in any outcome measure [8,15,17], which may justify the need for new long-term investigations.

Regarding weekly exercise frequency, the importance of 2 to 3 times a week for each muscle group in resistance exercises is observed in the American College of Sports Medicine (ACSM) guidelines [18]. Seven studies included this recommendation [9,11,12,13,15,16,17]. However, one study did not report the weekly exercise frequency of the intervention [14].

Regarding the duration of the sessions, some studies did not describe the length of the sessions. Of those that presented this information, the sessions lasted between 30 min and 2 h. The lack of such information raises doubts about the volume of the exercise and compromises possible replicability.

### 4.3. Participants

The heterogeneity of the study participants can be considered a positive aspect, since it allows a better interpretation of different contexts. However, this can also lead to possible errors when the results are not specified for each type of lower limb amputation or even for each sex and age. Another point to note is the small number of participants in each study. Only one study contained more than fifty participants [14]. Considering studies may present different levels of amputation, sex, and age, as well as a low number of participants, it is important to pay attention to potentially inaccurate results.

### 4.4. Exercise Protocols

The exercise protocols were varied and determined by the specific objective of the investigated studies. Unfortunately, many studies did not present fundamental details regarding the exercise protocols. It should be noted that the ACSM guidelines does not present guidelines for amputees, which may have resulted in a lack of standardized guidelines in the studies.

#### 4.4.1. Exercise Intensity

To calculate the intensity of a physical exercise, it is recommended to use the percentage calculation of 1RM, or another RM load, such as 10RM [5]. One study used the pre-test 10RM to define the 1RM load used in their intervention and then 50% of 1RM was used for the weightlifting circuit exercises [16]. According to the ACSM, exercises between 40–50% of 1RM can be beneficial for elderly and sedentary individuals [18]. 

In the case study [16], the individual was a 40-year-old man who was hypertensive, a smoker, and a bi-amputee. Due to the limitations above, exercises of 40–50% of 1RM may have been ideal in this case. Nevertheless, the exercise program contained closed chain exercises in addition to the weightlifting circuit, which were performed in three sets of ten repetitions with 50%, 75%, and 100% of 10RM. 

In another study [17], the intensity of 10RM was established, as they considered it safe for patients with lower limb amputations. 

Some studies did not report RM data but presented the load (kg) used in each exercise [10,15]. Furthermore, it should be noted that only these studies discussed exercise intensity with RM. This makes interpretation of the data difficult and also affects the study’s replicability.

#### 4.4.2. Number of Sets, Repetitions, and Rest Interval Length

Due to differences between the exercise protocols, the sets and repetitions were consequently different. There was also a lack of information about the sets and repetitions used in the studies. Rest interval lengths of two to three minutes between sets are recommended by the ACSM [18]. Few studies indicated the rest intervals used in the protocols, but they ranged from one to two minutes when provided [8,10].

#### 4.4.3. Type of Exercises

According to the ACSM [18], several models of resistance training and equipment can be used This review showed that different exercise models were used with a variety of equipment. Given the particularity of lower limb amputation, it may be necessary for exercises to be adapted so that they can be performed properly by an amputee [19,20]. Only a few studies [9,11] made clear that the researchers adapted the exercises for the participants. It is noteworthy that in addition to resistance training, the exercise protocols of some studies included other training modes, such as aerobic, flexibility, and balance exercises [9,10,11,14,15,16]. In addition, exercises for upper limbs were also used in some studies [16]. Furthermore, some investigations compared other exercise techniques with resistance training [11,14,15,17]. It is important to emphasize that using other exercise modes in the protocols can raise doubts about the effectiveness of resistance training in the interventions, making it impossible to affirm whether resistance training was the only type of activity that caused the outcomes reported in those investigations.

### 4.5. Resistance Training Effects

Strength deficit [3], balance-related adversities [21,22,23], changes in gait [3,24,25,26,27], and low back chronic pain [15,28] are some of the many problems that can accompany amputation. On this topic, the effects of resistance training on strength deficit, changes in gait, chronic low back pain, and adversities related to balance are reported below.

#### 4.5.1. Strength Gains

The strength deficit may be associated with lack of exercise and disuse of the prosthesis [3]. A study with amputee individuals demonstrated resistance training can reduce the strength deficit in this population [28], which is in line with some of the findings in this review [12,13,15,16,17]. Various exercise protocols using resistance training as the primary intervention led to strength gains in amputees [12,13,15,16,17].

In a bilateral amputee, there was a more significant strength gain in the hip of the transtibial limb compared to the transfemoral limb stem [15]. This outcome [3] corroborated the findings of another study reporting that transfemoral amputees have a more significant deficit in their hip muscles than transtibial amputees.

#### 4.5.2. Fall Risk and Balance Analyses

The increased risk of falling is related to a lack of balance [22,23]. As a result, the amputee becomes afraid of falling [21], increasing their distrust in their ability to balance. One study [9] demonstrated an improvement in dynamic balance during walking and an increase in balance confidence after an exercise protocol with resistance training and balance exercises. Likewise, another study used only resistance training as an intervention and reported improved balance confidence and attenuation of fear of falling [17].

In a resistance training intervention accompanied by dynamic balance exercises, there was an increase in balance in only one foot on an unstable surface, even without visual input and with imprecise somatosensory feedback [11]. However, unlike previous studies, no significant changes were seen in the activity-specific balance confidence test (ABC test) used to examine confidence in balance. There was also no relationship between postural control and confidence in balance.

#### 4.5.3. Gait and Muscle Changes

When comparing the effects of resistance training with proprioceptive neuromuscular facilitation technique, a significant difference in gait patterns was reported in amputees, such as in stride length and cadence [14]. In contrast, another study [8] found no significant changes in walking speed and ability in two children. These results may have been because of the low number of participants, and according to the authors, a possible underestimation of exercise and training frequency. However, there is no way to confirm this outcome since the study [8] lacked fundamental information about the exercise protocol. Finally, some investigations [10,17] demonstrated improvements in gait after training sessions. In addition, transtibial amputees had more considerable improvements than transfemoral amputees in relation to the gait pattern [9].

One study reported an increase in body weight in the non-amputated limb among amputees in the control group who maintained their normal activities [11], which can lead to postural asymmetries that can influence gait and cause low back pain and osteoarthritis [24]. This finding indicated that individuals who kept their normal activities maintained or worsened their gait pattern, unlike those who engaged in resistance training, demonstrating that resistance training may be essential to attenuate changes in walking in this population.

Amputee individuals consume more energy when walking than non-amputees [26]. However, after an exercise protocol in one study [15], there was a reduction in the oxygen consumption of individuals in the intervention group. This also occurred in a case study preparing an amputee for a bicycle race [15]. Nevertheless, it should be noted that in this latter investigation, the participant also exercised on a stationary bicycle in addition to resistance training. Regarding the ability of amputees to run, one study [15] found most transtibial and transfemoral amputees were able to run, but the bilateral amputee did not want to attempt running in this study.

#### 4.5.4. Chronic Low Back Pain

Many amputees suffer from low back pain [24], with one study reporting 46% of participants had chronic low back pain and 58% of these participants were not affected by this problem prior to amputation [29]. Only one investigation sought to identify the effects of resistance training in lower limb amputees with chronic low back pain [12]. Their findings demonstrated resistance training can contribute to strengthening essential muscles in the lumbar region. However, it should be noted that studies that attempted to understand changes in gait, attenuation of the strength deficit, and strengthening of the lumbar musculature after intervention with resistance training may contribute to the understanding of the topic of chronic low back pain, since changes in gait and strength deficit are factors associated with chronic low back pain [3,24,29].

### 4.6. Locomotion and Accessibility

It is known that people with disabilities suffer from a lack of accessibility. Some authors [30] reported one of the main difficulties in their research was the lack of accessible transportation for participants to travel to the intervention site. In line with this, one study [17] suggested that they could not obtain the results of their research if assistance for the transportation of participants was unavailable. In addition, another investigation [31] also presented transport as a barrier to the practice of resistance training by people with disabilities. They also mentioned that adaptations were necessary for the training space, equipment, and exercises. As this can be a determining factor for an intervention, it is essential to consider accessibility and transport issues.

One of the goals of resistance training is to facilitate the practitioner’s daily tasks [18]. In several studies presented in this review with walking and balance as an outcome measure, an improvement in these measures was reported after a resistance training protocol [9,10,15,17], including visual and somatosensory limitations [11]. In addition, another aspect to be highlighted is the study by Nolan [15] in which amputees could run after a training protocol.

### 4.7. Other Results

One study demonstrated cardiovascular improvement; however, resistance training was not the only exercise used in this study [16]. Changes in muscle tone occurred in another investigation [8].

Quality of life may be reduced after the amputation process [32,33]. Low back pain is one of the causes that affect the quality of life of amputees [3,34]. The findings of other studies [12] can contribute to the discussion of this problem, in addition to other investigations that can indirectly alleviate chronic low back pain. One study pointed to an improvement in the quality of life of people with disabilities; however, this study was not specifically conducted with amputees [30]. However, in contrast with this study [30], another study reported it was not possible to identify improvements in the amputee’s quality of life [10].

## 5. Review Limitations

The present work had some limitations. The varied study designs and other exercise methods, such as aerobic exercise, in addition to resistance training during the interventions should be highlighted. The short duration of the studies and lack of presented information regarding the number of sessions and their duration in some studies were also limiting factors. The low methodological quality of the studies should also be highlighted, including the lack of fundamental methodological details for future replicability of the studies. Another point that should be emphasized is the small sample size combined with the heterogeneity of the participants, making more solid analyses difficult.

## 6. Conclusions

This systematic review aimed to analyze the effects of resistance training in individuals with lower limb amputation by investigating whether resistance training for this population can provide benefits or has contraindications, as well as identify the main resistance training strategies for this population. Interventions with resistance training and other training methods were sufficient to achieve strength gains in muscles of the lower limbs, hips, core, and lumbar region. In addition, improved postural stability resulted in increased confidence in balance, improved gait pattern and speed when walking, as well as gaining the ability to run.

In light of the results reported in this review, it is worthwhile to note that resistance training combined with other exercises appears to be beneficial for this population, attenuating the strength deficit, risk of falling, changes in gait, and chronic low back pain. However, it is impossible to identify resistance training as the main factor responsible for these benefits from the findings in the investigations, nor even to indicate that the positive effects presented would be observed with this training method in isolation.

The benefits of resistance training in amputees, as well as in non-amputees, seem to outweigh the risks. Although unusual hazards were not found in this review, it is known that any training method can pose risks to healthy and unhealthy practitioners. It should be noted that it may be necessary to adapt resistance exercises to the condition of the lower limb amputee. Facilitating access to training can also assist in developing an effective exercise program.

Another point to consider is the need to develop new investigations, especially studies with higher methodological quality such as randomized clinical trials, preferably using only resistance training as an intervention to better understand the effects of the isolated intervention. As expected, interventions with resistance training made possible gains for this population. Given this, it is noteworthy that the main finding of this review is that the observed effects may differ according to the level of amputation, with mainly transtibial and transfemoral amputations analyzed.

## 7. Resistance Training Practical Recommendations for Amputees

Lower limb amputees can follow different exercise models, such as those using free weights, circuits, weight machines, and using their own body mass. However, closed chain exercises on weight machines should be prioritized. Other training modes, such as balance and aerobic exercises, should ideally be used to support resistance training. It should be noted that it is recommended that exercises be adapted to the practitioner if necessary. Training should be performed 2 to 3 times a week per muscle group, with sets varying between 1 and 3 for strength gains, with 10 to 12 or 15 repetitions, prioritizing the principle of progressivity. It was not possible to define an ideal rest period. Regarding the rest interval, we suggest following the ACMS recommendations (2014) of two to three minutes of rest.

It is recommended to use the 10RM test to define 1RM and the RM percentage should be defined based on the exercise program’s objectives. Thus, it is suggested to follow the ACSM’s recommendations (2014); however, one should consider starting an exercise program for this population with 40–50% of 1RM. Regarding the training objectives, strengthening the lumbar region, hips, and lower limbs is recommended to reduce the strength deficit. To alleviate chronic low back pain, strengthening the lumbar region is recommended. Finally, to improve balance and gait pattern, strengthening the muscles of the hips and lower limbs is recommended. Table 6 contain a Resistance training practical recommendations for amputees.

## Figures and Tables

**Figure 1 jfmk-08-00023-f001:**
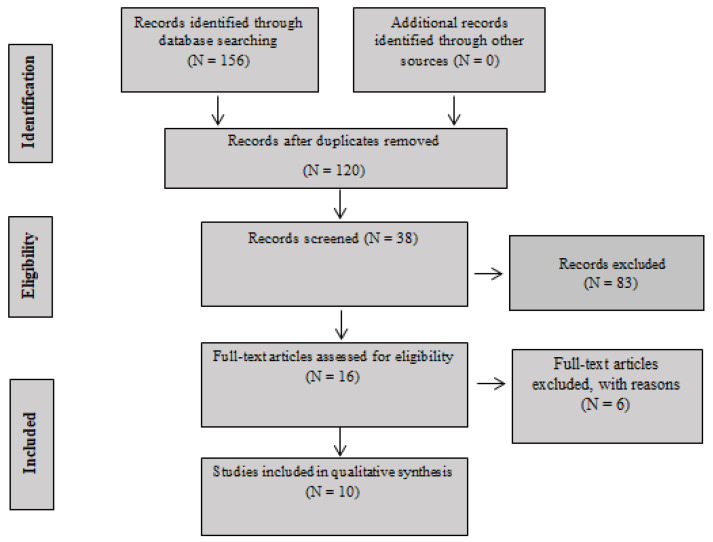
PRISMA Flow Diagram.

**Table 1 jfmk-08-00023-t001:** Used descriptors.

No	Descriptors
1	(“strength training” OR “exercise program” OR “strengthening Program” OR “resistance training” OR “exercise Prescription”) AND (amputee OR amputation) AND (“lower limb” OR “lower extremity”)

**Table 2 jfmk-08-00023-t002:** Found articles.

Database	Found Articles (N =156)
BVS	2
Cochrane	1
PubMed/Medline	21
Embase	56
Scopus	50
Web Of Science	26

**Table 3 jfmk-08-00023-t003:** TESTEX scale for analyzing the quality of studies.

Authors	Criteria												
	1	2	3	4	5	6	7	8	9	10	11	12	Total
García-García et al., 2021 [8]	N/A	N/A	N/A	N/A	N/A	2 **	N/A	N/A	N/A	N/A	1	0	3
Miller et al., 2017 [9]	1	N/A	N/A	N/A	N/A	2 **	N/A	N/A	1	N/A	1	0	5
Mosteiro-Losada et al., 2021 [10]	1	N/A	N/A	N/A	N/A	3	N/A	N/A	1	N/A	1	1	7
Schafer and Vanicek, 2021 [11]	1	1	1	1	0	1	1	2	1	0	1	0	10
Shin et al., 2018 [12]	1	N/A	N/A	N/A	N/A	1 *^,^**	0	N/A	1	N/A	0	0	3
Tipchatyotin et al., 2019 [13]	1	N/A	N/A	N/A	N/A	1 *^,^**	0	N/A	1	N/A	0	0	3
Anjum, Amjad, and Malik, 2016 [14]	1	1	0	1	0	0 *	N/A	2	1	1	0	0	7
Nolan, 2012 [15]	1	1	0	1	0	2	1	1	1	1	1	1	11
Donachy et al., 2004 [16]	N/A	N/A	N/A	N/A	N/A	2 **	N/A	N/A	N/A	N/A	1	1	4
Pauley, Devlin, and Madan-Sharma, 2014 [17]	1	1	1	1	1	3	1	2	1	1	0	1	14
García-García et al., 2021 [8]	N/A	N/A	N/A	N/A	N/A	2 **	N/A	N/A	N/A	N/A	1	0	3

* studies that did not report the number of dropouts, but all concluded; ** studies that did not report adverse effects; N/A not applicable; Eligibility criteria specified; 2 Randomization specified; 3 Allocation concealment; 4 Groups similar at baseline; 5 Blinding of assessor; 6 Outcome measures assessed in 85% of patients; 7 Intention-to-treat analysis; 8 Between-group statistical comparisons reported; 9 Point measures and measures of variability for all reported outcome measures; 10 Activity monitoring in control groups; 11 Relative exercise intensity remained constant; 12 Exercise volume and energy expenditure.

**Table 4 jfmk-08-00023-t004:** Study characteristics.

Author/Year/Country	Study Design/Aims	Participants	Amputation Level	Assessment Tools
García-García et al.2021, [8]; Spain	Case study. Provide information regarding th] characteristics and effectiveness of a rehabilitation exercise developed for children with lower-limb amputation.	N = 2; children. C1: boy, 8 years old. C2: girl, 9 years old	C1: Transtibial C2: Bilateral	Walking ability and walking speed; L-test of functional mobility; 10-m walk test (10MWT); tensiomyography (TMG).
Miller et al., 2017 [9]; USA	Non-randomized clinical trial. Explore the impact of a supervised community–based exercise program on balance, balance confidence, and gait in individuals with lower limb amputation	N = 16; mean age: 50.8 years (range 22–87); male (31.2%)/female (68.8%); amputation mean time: 8 years; prosthesis use mean time: 10.4 h/day; convenience sample	Not specified	Pre-test: PAR-Q; GAITRite Gold; figure-of-8 walk test (F8W); activity-specific balance confidence scale (ABC). Post-test: GAITRite Gold; figure-of-8 walk test (F8W); activity-specific balance confidence scale (ABC);
Mosteiro-Losada et al., 2021 [10]; Spain	Pilot study. Analyze functional mobility, walking speed, range of motion, and quality of life changes of lower limbs amputees after an exercise program.	N = 6; age: 56.83 ± 9.70 years; female (N = 1); male (N = 5);	Not specified	L-test; 10-m walk test (10MWT); range of motion; 36-item short form health survey (SF-36)
Schafer e Vanicek, 2021 [11]; United Kingdom	Randomized clinical trial. Evaluate the effectiveness of a 12-week personalized exercise program on postural control for individuals with lower limb amputation during different balancing conditions when the somatosensory, visual, and vestibular systems were challenged.	N = 14; control group (N = 7) male (N = 7), female (N = 0); age: 63 (DP = 17); amputation time: 18 years (SD = 21). Intervention group (N = 7) male (N = 4), female (N = 3); age: 60 years (SD = 12); amputation time: 10 years (SD = 17).	Transfemoral (N = 5) Transtibial (N = 2)	Sensory organization test (SOT); motor control test (MCT); ABC scale.
Shin et al., 2018 [12]; South Korea	Prospective study; Analyze the effect of lumbar strengthening exercise in lower-limb amputees with chronic low back pain	N = 19; mean age: 63.9 ± 7.4 years; amputation time: 39.6 ± 7.5.	Transfemoral (N = 5) Knee disarticulation (N = 1) Transtibial (N = 9) Syme (N = 1)	Visual analog scale (VAS); Korean version of the Oswestry Disability Index (K-ODI); Thomas test; Sorensen Test; trunk-raising test; prone-lying trunk-raising test.
Tipchatyotin et al., 2019 [13]; Thailand	A quasi-experimental study. Evaluate the effect of hip muscle strengthening exercise on gait performance in above-the-knee amputees.	N = 8; mean age: 52.5 ± 13.7 years; male (N = 6); female (N = 2);	Not specified	Gait parameters; 10-m walk test (10MWT); hip muscle strength
Anjum, Amjad, Malik, 2016 [14]; Pakistan	Randomized clinical trial. Determine the effects of proprioceptive neuromuscular facilitation (PNF) techniques as compared with traditional strength training (TPT) in improving ambulatory function in subjects with trans-tibial amputation.	N = 63; randomized groups: PNF (N = 31)/TPT (N = 32)	Transtibial	Locomotive capacity index and gait parameters
Nolan, 2012 [15]; Sweden	Randomized clinical trial. Investigate the effect of a 10-week training program on persons with lower limb amputation and determine if this training is sufficient to enable running.	N = 16; training group (N = 8): mean age 41.1 years (standard deviation (SD) = 8.4); average height 1.8 m (SD = 0.12); average body mass: 91.5 kg (SD = 25.5); amputation time: 8.2 years (SD = 9.2). Control group (N = 8): mean age: 49 years (SD = 9.1); average height: 1.7 m (SD = 0.08); average body mass: 76.2 kg (SD = 14.9); amputation time: 8.3 years (SD = 11.3);	Transtibial (N = 7) Transfemoral (N = 8) Bilateral (N = 1)	Hip strength; oxygen consumption; gait.
Donachy et al., 2004 [16]; USA	Case study. Describe the development of a strength and endurance training program designed to prepare an individual with left glenohumeral disarticulation and transtibial amputation for a bike trip across the USA.	N = 1; man; 40 years old	Left Transtibial amputation and left shoulder disarticulation	Timed sit-up test; 10RM test; test of peak VO_2_. Tests were modified to compensate for this individual’s characteristics.
Pauley, Devlin e Madan-Sharma, 2014 [17]; Canada	Randomized single-blind, crossover trial. Evaluate hip abductor strength training for transfemoral amputee patients.	N = 17; intervention group (N = 9); control group (N = 8); male (N = 13); female (N = 4); age: 67.8 years (SD = 5.2); amputation time: 7.3 years (SD = 8.2).	Right transfemoral amputation (N = 6) Left transfemoral amputation (N = 11)	Timed up and go test; two-minute walk test; hip abduction strength; ABC scale; Houghton scale.

**Table 5 jfmk-08-00023-t005:** Intervention characteristics and results.

**Author/Year/Country**	**Exercise Protocol Time**	**Interventions/Exercise Protocol**	**Results**
García-García et al., 2021 [8]; Spain	20 weeks; 1×/week; 2 h/day.	1st step: basic training: started with 6 exercises/20 s, progressed to 3 × 10 exercise, 2 min rest. 2nd step: coordination and lower-limb strengthening exercises: started with 1 × 10 exercises, after 3 × 10 exercises, 2 min rest.	Case 1: ↓ rectus femoris muscle tone; ↓ biceps femoris muscle tone; ↑ radial displacement velocity. Case 2: ↑ right rectus femoris muscle tone and ↓ transtibial rectus femoris muscle tone; No changes in biceps femoris; rectus femoris radial displacement velocity ↓ in limb with knee disarticulation and ↑ in transtibial limb. No significant change: biceps femoris radial displacement velocity. Walking ability and walking speed, the observed changes were of little relevance in both children.
Miller et al., 2017 [9]; USA	1 h/session; 2×/week; 6 weeks.	Stretching, core (trunk) and lower extremity strength and flexibility exercise; static and dynamic balance and gait activities. Exercise modifications and increased supervision were provided.	F8W test: ↑ dynamic balance ↓ dynamic balance. ABC test: ↑ 63.4% → 73.7%. ↑ balance confidence transtibial compared to transfemoral. Only 31% had an average score + 80%, compared to 25% pre-test. GAITRite: Comfortable walking speed: ↑average velocity (0.14 m/s), greater improvement in transtibial group; ↑ stride length in prosthetic and non-prosthetic side; ↑ cadence and single support. Fast walking speed: ↑ in all aspects mentioned in comfortable speed.
Mosteiro-Losada et al., 2021 [10]; Spain.	1st step: first w weeks; 1×/week 1 h/session; 2nd step: 3rd week; 1×/week; 3rd etapa: 4th week; 1×/week; 2 h/session. Warming (15 min); main (time n/i); calm down (10 min)	1st step: diaphragmatic breathing exercises; body awareness exercises. 2nd step: supine bridge. 3rd step: aimed at trunk stability and both upper and lower musculature. Participants with lower fitness levels: 10×/exercise w/20 s rest. Increase 1× repetition every 2 weeks. Participants with great fitness levels: 12×/exercises. Increase 1× repetition every week. Once all the participants were able to perform 15 repetitions of the proposed exercises, two circuit training workouts were proposed, including six exercises for 30 s each and a rest interval of 30 s between them.	All completed the study and there were no injuries. Significant improvements:↑ functional mobility (*p* = 0.007) and walking speed (*p* = 0.01); The training program did not have a significant impact on the participants’ range of motion or quality of life.
Schafer e Vanicek, 2021 [11]; United Kingdom	12 weeks; 2×/week circuit at university; 1×/week at home, 2×/week after 6 weeks	Intervention group: exercises included concentric and eccentric strengthening of key muscle groups (plantar flexors, knee extensors, hip extensors, flexors, abductors and adductors, and abdominal muscles) and dynamic balance (including picking up objects from the floor and balancing on a compliant surface). Control group: usual activities.	Intervention group: ↑ equilibrium score (*p* < 0.012, d = 1.45); no significant changes were observed for the other conditions; no significant changes in ABC score. Control group: ↑ weight in intact limb, causing asymmetry.
Shin et al., 2018 [12]; South Korea	8 weeks; 2×/week; 30 min/session	Lumbar strengthening exercises, lumbar stabilization exercises.	↑ abdominal muscle strength in comparison with a baseline (4.4 ± 0,7 → 4.8 ± 0.6); ↑ back extensor strength (2.6 ± 0.6 → 3.5 ± 1.2); ↑ back extensor endurance (22.3 ± 10,7 → 46.8 ± 35.1); ↓ Visual analog scale score (4.6 ± 2,2 → 2.6 ± 1.6); ↑ peak torque and flexors and extensors trunk total work.
Tipchatyotin et al., 2019 [13]; Thailand.	3 weeks; 2×/week	Isokinetic hip muscle training.	↑ hip strength and pelvic control during gait. Nonetheless, there was no significant change in gait speed, step length, and cadence.
Anjum, Amjad, Malik, 2016 [14]; Pakistan	4 weeks; 30 min/session.	PNF: weight bearing, weight shifting, balance exercise, single limb loading, stepping, and ST through sandbag. TPT: weight bearing, weight shifting, balance exercise, single limb loading, and stepping.	No significant difference: knee extension and flexion and hip extension. Significant difference: locomotive capacity index and gait parameter. PNF = 23.93 ± 4.24; ST = 18.18 ± 7.78 (*p* < 0.001)
Nolan, 2012 [15]; Sweden.	10 weeks; 2×/week w/1 day rest.	Intervention group: home training program with instructor: warm-up (20 min), balance and co-ordination exercises (5–10 min), hip strengthening exercises, cool-down (5–10 min). Hip strengthening exercises: slow and fast hip flexion and extension w/weight. Control group: continued with their usual activities (Nordic walking, swimming, aerobics, physiotherapy or no exercise at all).	No significant difference between intervention and control groups for height, weight, age, years as an amputee, strength, oxygen consumption in pre-test. ↓ intervention group body mass after training. Nonetheless, there was no significant difference for body mass between the two groups post-training; Strength results without bilateral amputee: ↑ 60° for the intact limb, all strength flexion and extension variables (with the exception of the intact limb extension peak force). For the residual limb, ↑ all strength variables. Bilateral strength results: appeared to exhibit strength differences between her transtibial and transfemoral limbs. Transtibial limb hip flexors appeared to be stronger than transfemoral limb flexors at both speeds. Hip extensor strength remained the same post-test. An increase in transtibial limb and transfemoral limb strength. Control group: no significant increase in strength in any of the members of the control group. However, intact limb peak extensor strength significantly reduced between pre and post-testing Training vs. control groups: not all differences in strength were found to be statistically significant. Intervention group: ↑ significant intact limb hip extensor strength compared with control group; ↑ significant residual limb hip flexor and extensor strength compared with control group. Oxygen consumption: ↓ oxygen consumption in intervention group. Running: Most transtibial amputees were able to run. All transfemoral amputee were able to run. The bilateral amputee did not want to run.
Donachy et al., 2004 [16]; USA	2 months; 1st step: 3×/week; 2nd step: 2×/week.	Weightlifting circuit: 1 min/1 min rest. Exercise: 6 upper limb exercises, 3 trunk exercises, and 4 lower limb exercises. 50% 1RM/10RM. Weight was increased as the subject’s strength increased. Lower limb weight training was limited to closed chain activities. ST: 3 × 10 50%, 75%, and 100% 10RM. Cycling: 20 min, 75% VO2peak. Core stability training.	↑ timed sit-up test (38 → 48); ↑ 36.8% leg press 10RM; ↑ other outcomes between 7.46% and 42.13%. ↑cardiovascular fitness
Pauley, Devlin e Madan-Sharma, 2014 [17]; Canada	8 weeks; 2×/week	Intervention group: ST and hip abductor. Control group: arm ergometer.	↑ 17% timed up and go test; ↑ 7% walking test; ↑ balance confidence; ↑ abductor strength (sitting or lying).

↑ improvement/gain/development; ↓ decrease/loss/reduction; ^N/I^ not informed; → to.

**Table 6 jfmk-08-00023-t006:** Resistance training practical recommendations for amputees.

Variable	Evidence-Based Recommendations
Muscle groups	Strength deficit reduction: lumbar region, hip musculature, and lower limbs. Chronic low back pain: lumbar region. Balance and gait: hip musculature and lower limbs.
Frequency	2× to 3× a week per muscle group.
Intensity	10RM test to define 1RM value. We suggest the ACSM’s recommendations (2014) consider starting training with 40–50% of 1RM.
Time	It was not possible to define an ideal time for resistance training.
Sets	Start with 1 series and gradually evolve to 3 series for strength gains in amputees.
Repetitions	Start with 10 repetitions, progressing to 12 and up to 15.
Rest	It was not possible to set an ideal rest interval, so we recommend following the ACSM’s recommendations of 2–3 min rest.
Progression	Ideally, there should be progressive weight, number of repetitions, and sets.
Type	It is recommended that resistance training be used with the support of other exercise modes, such as balance, flexibility, and aerobic exercises. Resistance training exercises: exercises that use free weights, weightlifting circuits, and use of own body mass. It is recommended that the exercises be adapted, if necessary. Should give preference to closed chain exercises on weight machines.
Equipment	It is recommended to adapt the equipment, if necessary. Weight machines, free weights, elastic bands, body weight, and ankle weights may be used.

## Data Availability

Not applicable.

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
