# Peer review of "Effects of Resistance Training in Individuals with Lower Limb Amputation: A Systematic Review"

_jfmk, 2023, doi:10.3390/jfmk8010023_

Round 1

Reviewer 1 Report

Dear Authors,

the aim of your manuscript is very interesting, since there is still poor knowledge about what kind of exercises can amputees make to improve their functional performances.

The Introduction section is very small, and very little focus was made on the possible effects of resistance training on amputess (or, at least, on what do you expect from resistance training protocols for amputess).

The Methodology of research is good, and the systematic search was correctly performed.

The Results section is fine.

Most of the issues I have to raise regard the Discussion section. 

Line 158: highest score in what?

Line 165-167: rephrase

Line 187: specify what ACSM stands for and use it futher in the manuscript (such as at line 207).

Lines 248-252; rephrase

Lines 259-261: it is not clear if you refer to the same patient who had different kind of lower limb amputations 

Line 288: increase in what?

Line 292: control group? Who are the individuals in the control group?

Lines 312-316: rephrase

Several English grammar mistakes (i.e. line 331, "reporting" instead of "reported", or "discuss" instead of "discussion" line 344) throughout the Discussion section

Lines 334-336: rephrase

The Discussion section need to be reviewed and improved.

Conclusions are fine.

I would not report the practical recommendations as a true section after conclusions, but as a separate file attached to the article.

Reviewer 2 Report

Thank you for the good research. Overall, editorial fixes and table readability improvements are needed. I hope the following comments will make the paper better.

Round 2

Reviewer 1 Report

Lines 172-174: In a bilateral amputee, there was a more significant strength gain in the hip of the transtibial limb compared of his transfemoral limb stem. They corroborate the findings of another study, who indicated that transfemoral amputees have a more significant deficit in the hip muscles than transtibial amputees.

Please rephrase like this: "In one study [put the reference], a bilateral amputee was found to have more significant strenght" etc. Furthermore, change that "they" with "This outcome".

Line 304: in relation to the gait. Please change with "in relation to the gait pattern"

Lines 307-311: One study reported an increase in mass in the non-amputated limb among amputees in the control group who maintained their normal activities, which can lead to postural asymmetries that can influence gait and cause low back pain and osteoarthritis.

What do you mean with "mass"? Because if you refer to muscle mass, it is not clear why an increase in muscle mass should lead to postural asymmetries worsening the gait pattern. Please make this sentence clearer.

Lines 330-334: correct as follows: "However, it should be noticed that studies that attempted to understand changes in gait, attenuation of the strength deficit, and strengthening of the lumbar musculature after an intervention with resistance training can contribute to the understanding of the topic of chronic low back pain, since changes in gait and strength deficit are factors associated with chronic low back pain.

Please make these corrections, then the manuscript can be considered suitable for publication.

.
